# Learned Region Sparsity and Diversity
# Also Predict Visual Attention

**Zijun Wei**[1*], **Hossein Adeli**[2*], **Gregory Zelinsky**[1,2], **Minh Hoai**[1], **Dimitris Samaras**[1]

1. Department of Computer Science   2. Department of Psychology – Stony Brook University
1.{zijwei, minhhoai, samaras}@cs.stonybrook.edu
2.{hossein.adelijelodar, gregory.zelinsky}@stonybrook.edu
*. Both authors contributed equally to this work

## Abstract

Learned region sparsity has achieved state-of-the-art performance in classification tasks by exploiting and integrating a sparse set of local information into global decisions. The underlying mechanism resembles how people sample information from an image with their eye movements when making similar decisions. In this paper we incorporate the biologically plausible mechanism of Inhibition of Return into the learned region sparsity model, thereby imposing diversity on the selected regions. We investigate how these mechanisms of sparsity and diversity relate to visual attention by testing our model on three different types of visual search tasks. We report state-of-the-art results in predicting the locations of human gaze fixations, even though our model is trained only on image-level labels without object location annotations. Notably, the classification performance of the extended model remains the same as the original. This work suggests a new computational perspective on visual attention mechanisms, and shows how the inclusion of attention-based mechanisms can improve computer vision techniques.

## 1   Introduction

Visual spatial attention refers to the narrowing of processing in the brain to particular objects in particular locations so as to mediate everyday tasks. A widely used paradigm for studying visual spatial attention is visual search, where a desired object must be located and recognized in a typically cluttered environment. Visual search is accompanied by observable estimates—in the form of gaze fixations—of how attention samples information from a scene while searching for a target. Efficient visual search requires prioritizing the locations of features of the target object class over features at locations offering less evidence for the target [31]. Computational models of visual search typically estimate and plot goal directed prioritization of visual space as *priority maps* for directing attention [32]. This form of target directed prioritization is different from the *saliency* modeling literature, where bottom-up feature contrast in an image is used to predict fixation behavior during the free-viewing of scenes [16].

The field of fixation prediction is highly active and growing [2], although it was not until fairly recently that attention researchers have begun using the sophisticated object detection techniques developed in the computer vision literature [8, 18, 31]. The dominant method used in the visual search literature to generate priority maps for detection has been the exhaustive detection mechanism [8, 18]. Using this method, an object detector is applied to an image to provide bounding boxes that are then combined, weighted by their detection scores, to generate a priority map [8]. While these models have had success in predicting behavior, training these detectors requires human labeled bounding boxes, which are expensive and laborious to collect, and also prone to individual annotator differences.

An alternative approach to modeling visual attention is to determine how model and behavioral task performance depends on shared core computational principles [24]. To this end, a new class of attention-inspired models have been developed and applied to tasks ranging from image captioning [30] to hand writing generation [13], where selective spatial attention mechanisms have been shown to emerge [1, 25]. By requiring visual inputs to be gated in a manner similar to the human gating of visual inputs via fixations, these models are able to localize or "attend" selectively to the most informative regions of an input image while ignoring irrelevant visual inputs [25, 1]. This built in attention mechanism enables the model of [30], trained only on generating captions, to bias the visual input so as to gate only relevant information when generating each word to describe an image. Priority maps were then generated to show the mapping of attended image areas to generated words. While these new models show attention-like behavior, to our knowledge none have been used to predict actual human allocations of attention.

The current work bridges the behavioral and computer vision literatures by using a classification model that has biologically plausible constraints to create a priority map for the purpose of predicting the allocation of spatial attention as measured by changes in fixation. The specific image-category classification model that we use is called Region Ranking SVM (RRSVM) [29]. This model was developed in our recent work [29], and it achieved state-of-the-art performance on a number of classification tasks by learning categorization with locally-pooled information from input images. This model works by imposing sparsity on selected image areas that contribute to the classification decision, much like how humans prioritize visual space and sample with fixations only a sparse set of image locations while attempting to detect and recognize object categories [4]. We believe that this analogy between sparse sampling and attention makes this model a natural candidate for predicting attention behavior in visual search tasks. It is worth noting that this model was originally created for object classification and not localization, hence no object localization data is used to train it, unlike standard fixation prediction algorithms [16, 17].

There are two contributions of our work. First, we show that the RSSVM model approaches state-of-the-art in predicting the fixations made by humans searching for the same targets in the same images. This means that a model trained solely for the purpose of image classification, without any localization data, is also able to predict the locations of fixations that people make while searching for the to-be-classified objects. Second, we incorporate the biologically plausible constraint of Inhibition of Return [10], which we model by requiring a set of diverse (minimally overlapping) sparse regions in RRSVM. Incorporating this constraint, we are able to reduce the error in fixation prediction (up to 21%). Importantly, adding the Inhibition of Return constraint does not affect the classification performance. By building this bridge, we hope to show how automated object detection might be improved by the inclusion of an attention mechanism, and how a recent attention-inspired approach from computer vision might illuminate how the brain prioritizes visual information for the efficient direction of spatial attention.

## 2   Region Ranking SVM

Here we review Region Ranking SVM (RRSVM) [29]. The main problem addressed by RRSVM is image classification, which aims to recognize the semantic category of an image, such as whether the image contains a certain object (e.g., car, cat) or portrays a certain action (e.g., jumping, typing). RRSVM evaluates multiple local regions of an image, and subsequently outputs the classification decision based on a sparse set of regions. This mechanism is noteworthy and different from other approaches that aggregate information from multiple regions indistinguishably (e.g., [23, 28, 22, 14]).

RRSVM assumes training data consisting of images $\{\mathbf{B}_i\}_{i=1}^n$ and associated binary labels $\{y_i\}_{i=1}^n$ indicating the presence or absence of the visual element (object or action) of interest. To account for the uncertainty of each semantic region in an image, RRSVM considers multiple local regions. The number of regions can differ between images, but for brevity, assume each image has the same number of regions. Let $m$ be the number of regions for each image, and $d$ the dimension of each region descriptor. RRSVM represents each image as a matrix $\mathbf{B}_i \in \Re^{d \times m}$, but the order of the columns can be arbitrary. RRSVM jointly learns a region evaluation function and a region selection function by minimizing: $\lambda ||\mathbf{w}||^2 + \sum_{i=1}^n (\mathbf{w}^T \Gamma(\mathbf{B}_i; \mathbf{w})\mathbf{s} + b - y_i)^2$ subject to the constraints: $s_1 \geq s_2 \geq \cdots \geq s_m \geq 0$ and $h(\Gamma(\mathbf{B}_i; \mathbf{w})\mathbf{s}) \leq 1$. Here $h(\cdot)$ is the function that measures the spread of the column vectors of a matrix: $h([\mathbf{x}_1, \cdots, \mathbf{x}_n]) = \sum_{i=1}^n \left|\left| \mathbf{x}_i - \frac{1}{n} \sum_{i=1}^n \mathbf{x}_i \right|\right|^2$. $\mathbf{w}$ and $b$ are the weight vector and the bias term of an SVM classifier, which are the parameters of the region

evaluation function. $\Gamma(\mathbf{B};\mathbf{w})$ denotes a matrix that can be obtained by rearranging the columns of the matrix $\mathbf{B}$ so that $\mathbf{w}^T\Gamma(\mathbf{B};\mathbf{w})$ is a sequence of non-increasing values. The vector $\mathbf{s}$ is the weight vector for combining the SVM region scores for each image [15]; this vector is common to all images of a class.

The objective of the above formulation consists of the regularization term $\lambda||\mathbf{w}||^2$ and the sum of squared losses. This objective is based purely on classification performance. However, note that the classification decision is based on both the region evaluation function (i.e., $\mathbf{w}, b$) and the region selection function (i.e., $\mathbf{s}$), which are simultaneously learned using the above formulation. What is interesting is that the obtained $\mathbf{s}$ vector is always sparse. An experiment [29] on the ImageNet dataset [27] with 1000 classes showed that RRSVM generally uses 20 regions or less (from hundreds of local regions considered). This intriguing fact prompted us to consider the connection between sparse region selection and visual attention. Would machine-based discriminative localization reflect the allocation of human attention in visual search? It turns out that there is compelling evidence for a relationship, as will be shown in the experiment section. This relationship can be strengthened if RRSVM is extended to incorporate *Inhibition of Return* in the region selection process, which will be explained next.

## 3 Incorporating Inhibition of Return into Region Ranking SVM

A mechanism critical to the modeling of human visual search behavior is Inhibition of Return: the lower probability of re-fixating on or near already attended areas, possibly mediated by lateral inhibition [16, 20]. This mechanism, however, is not currently enforced in the formulation of RRSVM, and indeed the spatial relationship between selected regions is not considered. RRSVM usually selects a sparse set of regions, but the selected regions are free to overlap and concentrate on a single image area.

Inspired by Inhibition of Return, we consider an extension of RRSVM where non-maxima suppression is incorporated into the process of selecting regions. This mechanism will select the local maximum for nearby activation areas (a potential fixation location) and discard the rest (non-maxima nearby locations). The biological plausibility of non-maxima suppression has been discussed in previous work, where it was shown to be a plausible method for allowing the stronger activations to stand out (see [21, 7] for details).

To incorporate non-maxima suppression in the framework of RRSVM, we replaced the region ranking procedure $\Gamma(\mathbf{B};\mathbf{w})$ of RRSVM by $\Psi(\mathbf{B}_i;\mathbf{w},\alpha)$, a procedure that ranks and subsequently returns the list of regions that do not significantly overlap with one another. In particular, we use intersection over union to measure overlap, where $\alpha$ is a threshold for tolerable overlap (we set $\alpha = 0.5$ in our experiments). This leads to the following optimization problem:

$$\underset{\mathbf{w},\mathbf{s},b}{\text{minimize}} \; \lambda||\mathbf{w}||^2 + \sum_{i=1}^{n}(\mathbf{w}^T\Psi(\mathbf{B}_i;\mathbf{w},\alpha)\mathbf{s} + b - y_i)^2 \tag{1}$$

$$\text{s.t. } s_1 \geq s_2 \geq \cdots \geq s_m \geq 0, \tag{2}$$

$$h(\Psi(\mathbf{B}_i;\mathbf{w},\alpha)\mathbf{s}) \leq 1. \tag{3}$$

The above formulation can be optimized in the same way as RRSVM in [29]. It will yield a classifier that makes a decision based on a sparse and diverse set of regions. Sparsity is inherited from RRSVM, and location diversity is attained using non-maxima suppression. Hereafter, we refer to this method as Sparse Diverse Regions (SDR) classifier.

## 4 Experiments and Analysis

We present here empirical evidence showing that learned region sparsity and diversity can also predict visual attention. We first describe the implementation details of RRSVM and SDR. We then consider attention prediction under three conditions: (1) single-target present, that is to find the one instance of a target category appearing in a stimulus image; (2) target absent, i.e., searching for a target category that does not appear in the image; and (3) multiple-targets present, i.e., searching for multiple object categories where at least one is present in the image. Experiments are performed on three datasets POET [26], PET [11] and MIT900 [8], which are the only available datasets for object search tasks.

## 4.1 Implementation details of RRSVM and SDR

Our implementation of RRSVM and SDR is similar to [29], but we consider more local regions. This yields a finer localization map without changing the classification performance. As in [29], the feature extraction pipeline is based on VGG16 [28]. The last fully connected layer of VGG16 is removed and the remaining fully connected layer is converted to a fully convolutional layer. To compute feature vectors for multiple regions of an image, the image is resized and then fed into VGG16 to yield a feature map with 4096 channels. The size of the feature map depends on the size of the resized image, and each feature map corresponds to a subwindow of the original image. By resizing the original image to multiple sizes, one can compute feature vectors for multiple regions of the original image. In this work, we consider 7 different image sizes instead of the three sizes used by [28, 29]. The first three resized images are obtained by scaling the image isotropically so that the smallest dimension is 256, 384, or 512. For brevity, assuming the width is smaller than the height, this yields three images with dimensions $256 \times a, 384 \times b$, and $512 \times c$. We consider four other resized images with dimensions $256 \times b, 384 \times c, 384 \times a, 512 \times b$. These image sizes correspond to local regions having an aspect ratio of either 2:3 or 3:2, while the isotropically resized images yield square local regions. Additionally, we also consider horizontal flips of the resized images. Overall, this process yields 700 to 1000 feature vectors, each corresponding to a local image region.

The RRSVM and SDR classifiers used in the following experiments are trained on the trainval set of PASCAL VOC 2007 dataset [9] unless otherwise stated. This dataset is distinct from the datasets used for evaluation. For SDR, the non-maxima suppression threshold is 0.5, and we only keep the top ranked regions that have non-zero region scores ($s_i \geq 0.01$). To generate a priority map, we first associate each pixel with an integer indicating the total number of selected regions covering that pixel, then apply a Gaussian blur kernel to the integer valued map, with the kernel width tuned on the validation set.

To test whether learned region sparsity and diversity predicts human attention, we compare the generated priority maps with the behaviorally-derived fixation density maps. To make this comparison we use the Area Under the ROC Curve (AUC), a commonly used metric for visual search task evaluation [6]. We use the publicly available implementation of the AUC evaluation from the MIT saliency benchmark [5], specifically the AUC-Judd implementation for its better approximation.

## 4.2 Single-target present condition

We consider visual attention in the single-target present condition using the POET dataset [26]. This dataset is a subset of PASCAL VOC 2012 dataset [9], and it has 6270 images from 10 object categories (aeroplane, boat, bike, motorbike, cat, dog, horse, cow, sofa and dining table). The task was two-alternative forced choice for object categories, approximating visual search, and eye movement data were collected from 5 subjects as they freely viewed these images. On average, 5.7 fixations were made per image. The SDR classifier is trained on the trainval set of PASCAL VOC 2007 dataset, which does not overlap with the POET dataset. We randomly selected one third of the images for each category to compile a validation set for tuning the width of the Gaussian blur kernel for all categories. The rest were used as test images.

For each test image, we compare the priority map generated for the selected regions by RRSVM with the human fixation density map. The overall correlation is high, yielding a mean AUC score of 0.81 (on all images of 10 object classes). This is intriguing because RRSVM is optimized for classification performance only; joint classification is apparently related to discriminative localization by human attention in the context of a visual search task. By incorporating Inhibition of Return into RRSVM, we observe even stronger correlation with human behavior, with the mean AUC score obtained by SDR now being 0.85.

The left part of Table 1 shows AUC scores for individual categories of the POET dataset. We compare the performance of other attention prediction baselines. All recent fixation prediction models [8, 19, 31] apply object category detectors on the input image and combine the detection results to create priority maps. Unfortunately, direct comparison to these models is not currently possible due to the unavailability of needed code and datasets. However, our RCNN [12] baseline, which is the state-of-the-art object detector on this dataset, should improve the pipelines of these models. To account for possible localization errors and multiple object instances, we keep all the detections with a detection score greater than a threshold. This threshold is chosen to maximize the

Table 1: AUC scores on POET and PET test sets

| Model | POET | | | | | | | | | | | PET |
| | aero | bike | boat | cat | cow | table | dog | horse | mbike | sofa | **mean** | multi-target |
|---|---|---|---|---|---|---|---|---|---|---|---|---|
| SDR | **0.87** | **0.85** | **0.83** | **0.89** | **0.88** | **0.79** | **0.88** | **0.86** | 0.86 | **0.77** | **0.85** | **0.83** |
| RCNN | 0.84 | 0.83 | 0.79 | 0.84 | 0.81 | 0.76 | 0.83 | 0.80 | **0.87** | 0.76 | 0.82 | 0.77 |
| CAM [34] | 0.86 | 0.78 | 0.78 | 0.88 | 0.84 | 0.74 | 0.87 | 0.84 | 0.83 | 0.67 | 0.82 | 0.65 |
| AnnoBoxes | 0.85 | 0.86 | 0.81 | 0.84 | 0.84 | 0.79 | 0.80 | 0.80 | 0.88 | 0.80 | 0.83 | 0.82 |

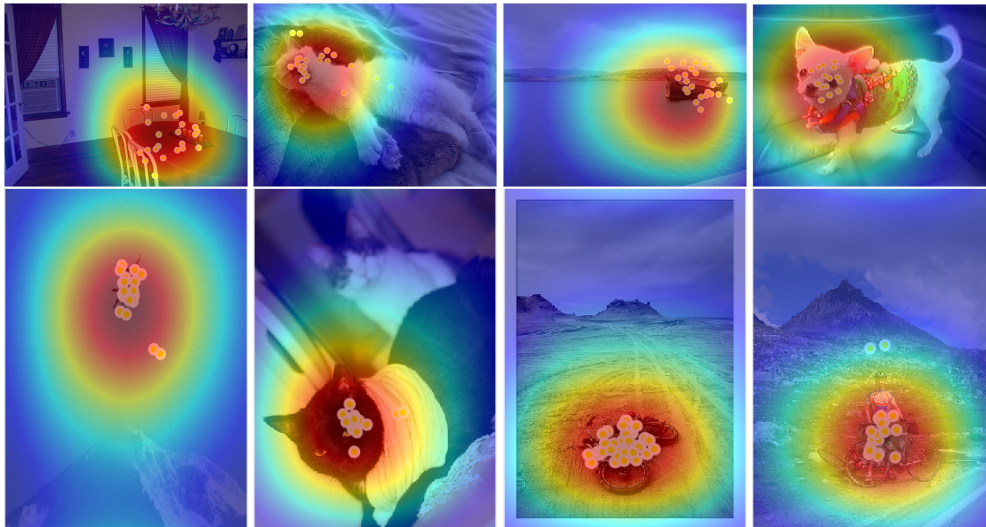

Figure 1: **Priority maps generated for SDR on the POET dataset.** Warm colors represent high values. Dots represents human fixations. Best viewed on a digital device.

detector's F1 score, which is the harmonic mean between precision and recall. We also consider a variant method where only the top detection is kept, but the result is not as good. We also consider the recently proposed weakly-supervised object localization approach of [34], which is denoted as CAM in Table 1. We use the released model to extract features and train a linear SVM on top of the features. For each test image, we weigh a linear sum of local activations to create an activation map. We normalize the activation map to get the priority map. We even compare SDR with a method that directly uses the annotated object bounding boxes to predict human attention, which is denoted as AnnoBoxes in the table. For this method, the priority map is created by applying a Gaussian filter to a binary map where the center of the bounding box over the target(s) is set to 1 and everywhere else 0. Notably, the methods selected for comparison are strong models for predicting human attention. RCNN has an unfair advantage over SDR because it has access to localized annotations in its training data, and AnnoBoxes even assumes the availability of object bounding boxes for test data. As can be seen from Table 1, SDR significantly outperforms the other methods. This provides strong empirical evidence suggesting that learned region sparsity and diversity is highly predictive human attention. Fig. 1 shows some randomly selected results from SDR on test images.

Note that the incorporation of Inhibition of Return into RRSVM and the consideration of more local regions does not affect the classification performance. When evaluated on the PASCAL VOC 2007 test set, the RRSVM method that uses local regions corresponding to 3 image scales (as in [29]), the RRSVM method that uses more regions with different aspect ratios (as explained in Sec. 4.1), and the RRSVM method that incorporates the NMS mechanism (i.e., SDR), all achieve a mean AP of 92.9%. SDR, however, is significantly better than RRSVM in predicting fixations during search tasks, increasing the mean AUC score from 0.81 to 0.85. Also note that the predictive power of SDR is not sensitive to the value of $\alpha$: for aeroplane on the POET dataset, the AUC scores remain the same (0.87) when $\alpha$ is varied from 0.5 to 0.7.

Figure 2 shows some examples highlighting the difference between the regions selected by RRSVM and SDR. As can be seen, incorporating non-maxima suppression encourages greater dispersion of

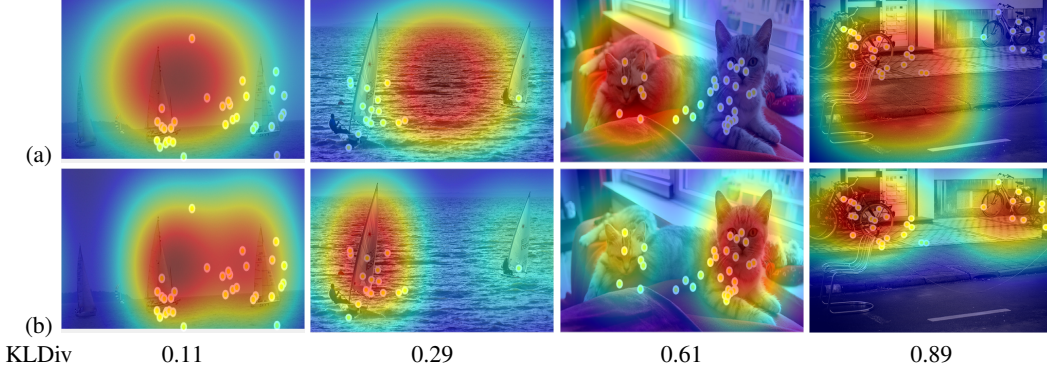

| | | | |
|---|---|---|---|
| (a) | | | |
| (b) | | | |
| KLDiv | 0.11 | 0.29 | 0.61 | 0.89 |

Figure 2: **Comparison between RRSVM and SDR on the POET dataset.** (a): priority maps created by RRSVM, (b): priority maps generated by SDR. SDR better captures fixations when there are multiple instances of the target categories. The KL Divergence scores between RRSVM and SDR are reported in the bottom row.

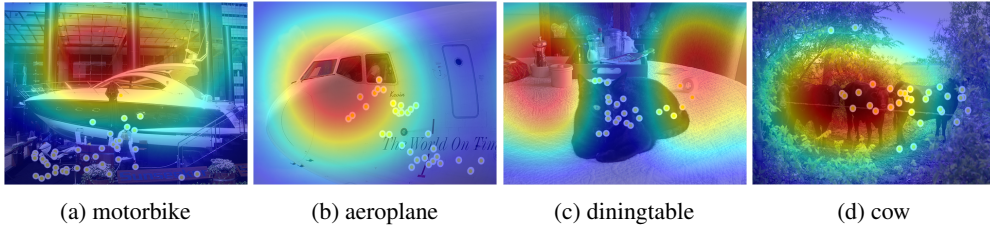

(a) motorbike          (b) aeroplane          (c) diningtable          (d) cow

Figure 3: **Failure cases**. Representative images where the priority maps produced by SDR are significantly different from human fixations. The caption under each image indicates the target category. The modes of failure are: (a) failure in classification; (b) and (c) existence of a more attractive object (text or face); (d) co-occurrence of multiple objects. Best viewed on digital devices.

the sparse areas as opposed to a more clustered distribution in RRSVM. This in turn better predicts attention when there are multiple instances of the target object in the display.

Figure 3 shows representative cases where the priority maps produced by SDR are significantly different from human fixations. The common failure modes are: (1) failure to locate the correct region for correct classification (see Fig 3a); (2) particularly distracting elements in the scene, such as text (3b) or faces (3c); (3) failure to attend to multiple instances of the target categories. Tuning SDR using human fixation behavioral data [17] and combining SDR with multiple sources of guidance information [8], including saliency and scene context, could mitigate some of the model limitations.

### 4.3 Target absent condition

To test whether SDR is able to predict people's fixations when the search target is absent, we performed experiments on 456 target-absent images from the MIT900 dataset [8]. Human observers were asked to search for people in real world scenes. Eye movement data were collected from 14 searchers who made roughly 6 fixations per image, on average. We picked a random subset of 150 images to tune the Gaussian blur parameter and reported the results for the remaining 306 images. We noticed that the sizes and poses of the people in these images were very different from those of the training samples in VOC2007, which could have led to poor SDR classification performance. In order to address this issue, we augmented the training set of SDR with 456 images from MIT900 that contain people. The added training examples were a disjoint set from the target-absent images for evaluation.

On these target absent cases, SDR achieves an AUC score of 0.78. As a reference, the method of Ehinger et al. [8] also achieves AUC of 0.78. But the two methods are not directly comparable because Ehinger et al. [8] used a HOG-based person detector that was trained on a much larger dataset with location annotation.

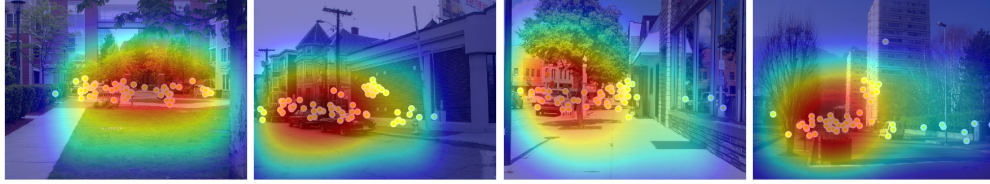

Figure 4: **Priority map predictions using SDR on some MIT target-absent stimuli.** Warm colors represent high probabilities. Dots indicate human fixations. Best viewed on a digital device.

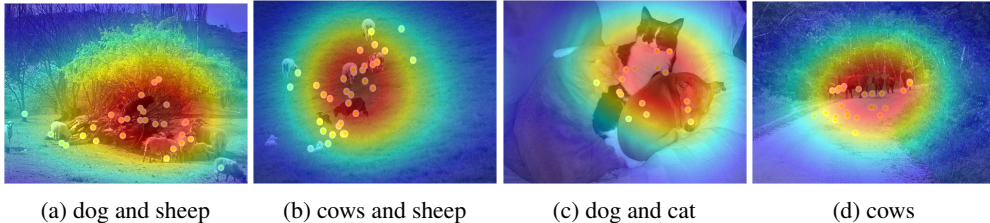

|          (a) dog and sheep          |          (b) cows and sheep          |          (c) dog and cat          |          (d) cows          |

Figure 5: **Visualization of SDR prediction on the PET dataset**. Note that the high classification accuracy ensures that more reliable regions are detected.

Figure 4 shows some randomly selected results from the test set demonstrating SDR's success in predicting where people attend. Interestingly, SDR looks at regions that either contain person-like objects or are likely to contain persons (e.g., sidewalks), with the latter observation likely the result of sidewalks co-occurring with persons in the positive training samples (a form of scene context effect).

### 4.4 Multiple-target attention

We considered human visual search behavior when there were multiple targets. The experiments were performed on the PET dataset [11]. This dataset is a subset of PASCAL VOC2012 dataset [9], and it contains 4135 images from 6 animal categories (cat, dog, bird, horse cow and sheep). Four subjects were instructed to find **all** of the animals in each image. Eye movements were recorded, where each subject made roughly 6 fixations per image. We excluded the images that contained people to avoid ambiguity with the animal category. We also removed the images that were shared with the PASCAL VOC 2007 dataset to ensure no overlap between training and testing data. This yielded a total of 3309 images from which a random set of 1300 images were selected for tuning the Gaussian kernel width parameter. The remaining 2309 images were used for testing.

To model the search for multiple categories in an image, for all methods except AnnoBoxes we applied six animal classifiers/detectors simultaneously to the test image. For each classifier/detector of each category, a threshold was selected to achieve the highest $F_1$ score on the validation data. The prediction results are shown in the right part of Tab. 1. SDR significantly outperforms other methods. Notably, CAM performs poorly on this dataset, due perhaps to the low classification accuracy of that model (83% mAP on VOC 2007 test set as opposed to 93% of SDR). Some randomly selected results are shown in Fig. 5.

### 4.5 Center Bias

For the POET dataset, some of the target objects are quite iconic and in the center of the image. For these cases, a simple center bias map might be a good predictor of the fixations. To test this, we generated priority maps by setting the center of the image to 1 and everywhere else 0, and then applying a Gaussian filter with sigma tuned on the validation set. This simple Center Bias (CB) map achieved an AUC score of 0.84, which is even higher than some of the methods presented in Tab. 1. This prompted us to analyze whether the good performance of SDR is simply due to center bias.

An intuitive way to address the CB problem would be to use Shuffled AUC (sAUC) [33]. However, sAUC favors true positives over false negatives and gives more credit to off-center information [3], which may lead to biased results. This is especially true when the datasets are center-biased. The sAUC scores for RCNN, AnnoBox, CAM, SDR, and Inter-Observer [3] are 0.61, 0.61, 0.65, 0.64, and 0.70, respectively. SDR outperforms AnnoBox and RCNN by 3% and is on par with CAM. Also

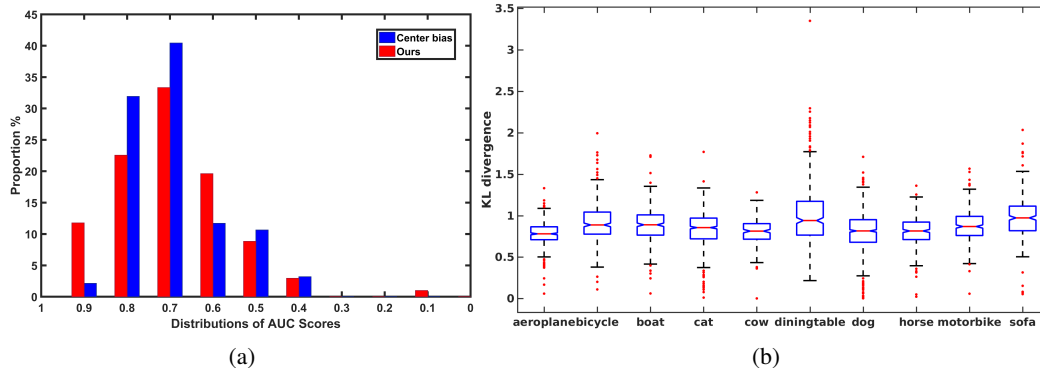

(a)                                         (b)

Figure 6: (a): Red bars: the distribution of AUC scores of SDR for which the AUC scores of Center Bias are under 0.6. Blue bars: the distribution of AUC scores Center Bias where AUC scores of SDR are under 0.6. (b): The box plot for the distributions of KL divergence between Center Bias and SDR scores on each class in POET dataset. The KL divergence distribution revealed that the priority maps created by Center Bias are significantly different from the ones created by SDR.

note that sAUC for Inter-Observer is 0.70, which suggests the existence of center bias in POET (the sAUC score of Inter-Observer on MIT300 [17] is 0.81) and raises a concern that sAUC might be misleading for model comparison using this dataset.

To further address the concern of center bias, we show in Fig. 6 that the priority maps produced by SDR and Center Bias are quite different. Fig. 6a plots the distribution of the AUC scores for one method when the AUC scores of the other method was low ($< 0.6$). The spread of these distributions indicate a low correlation between the errors of the two methods. Fig. 6b shows a box plot of the distribution of KL divergence [6] between the priority maps generated by SDR and Center Bias. For each category, the mean KL divergence value is high, indicating a large difference between SDR and Center Bias. For a more qualitative intuition of KL divergence in these distributions, see Figure 2.

The center bias effect in PET and MIT900 is not as pronounced as in POET because there are multiple target objects in the PET images and the target objects in the MIT900 dataset are relatively small. For these datasets, Center Bias achieves AUC scores of 0.78 and 0.72, respectively. These numbers are significantly lower than the results obtained by SDR, which are 0.82 and 0.78, respectively.

## 5   Conclusions and Future Work

We introduced a classification model based on sparse and diverse region ranking and selection, which is trained only on image level annotations. We then provided experimental evidence from visual search tasks under three different conditions to support our hypothesis that these computational mechanisms might be analogous to computations underlying visual attention processes in the brain.

While this work is not the first to use computer vision models to predict where humans look in visual search tasks, it is the first to show that core mechanisms driving high model performance in a search task also predict how humans allocate their attention in the same tasks. By improving upon these core computational principles, and perhaps by incorporating new ones suggested by attention mechanisms, our hope is to shed more light on human visual processing.

There are several directions for future work. The first is to create a visual search dataset that mitigates the center bias effect and avoids cases of trivially easy search. The second is to incorporate into the current model known factors affecting search, such as a center bias, bottom-up saliency, scene context, etc., to better predict shifts in human spatial attention.

**Acknowledgment.** This project was partially supported by the National Science Foundation Awards IIS-1161876 and IIS-1566248 and the Subsample project from the Digiteo Institute, France.

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
