[Reviews · NeurIPS 2016]

Reviewer 1

Summary

The paper proposes a Sparse Diverse Region Classifier (SDR), a method based on Region Ranking SVM (RRSVM) [29], which imposes diversity on the regions selected by RRSVM when training an image classifier. Diversity in the selected regions is introduced by a mechanism called "Inhibition of Return", which is in charge of only selecting regions with low spatial overlap. This overlap is computed using the traditional intersection over union criterion used in non-maximun suppresion (NMS). Experiments are conducted on the POET[26], MIT900 [8], PET [13] datasets, using a RRSVM and SDR models trained on the Pascal VOC 2007 dataset. The evaluation covers different scenarios: a) Single-target is present, b) the target is absent, and c) when multiple-targets are present. The Judd variation of the Area Under Curve (AUC-Judd) is used as a performance metric, where the ability of the method to recover visual fixation points is measured. Experimental results show that the introduced Inhibition Return mechanism introduced via NMS, brings a boost in performance of ~4 performance.

Qualitative Assessment

- The content of the paper is well presented and accompanied by clear illustrations which make the content of the paper easy to follow. Likewise, the experimental protocol followed in the paper is quite clear, which eases reproducibility of the results obtained in this work. Having said this, I consider there are two major flaws in the paper: - I appreciate the comparison wrt. to methods that use the combination of the output of object detectors to create the priority maps. However, in my opinion, despite the fact that R-CNN takes into account object annotations during training, it is not a strong baseline as is claimed in the manuscript due to the following reasons. First, R-CNN uses which has the weakness of using Selective Search (Uijlings et al., IJCV'13) which is a generic method for generating object proposals. R-CNN is not anymore the state-of-the-art, I recommend repeating the experiment with Faster-RCNN, Ren et al. CVPR'15 which addresses that weakness. Second, and more critical, R-CNN is trained to optimizing the object detection which is measured by the area under the curve where matching between predicted object regions and ground-truth is computed by taking into account object bounding boxes (not the fixation points). Since these are two different tasks, ie. object detection (localizing the full extent of the object) VS. human visual attention (localizing the specific regions that define the object), in my opinion it is not surprising that R-CNN has sub-standard performance. - Regarding the AnnoBoxes baselines, I agree that it is exploiting more information (annotated bounding boxes on test images). However, I believe that the way in which this information is used, reduces significantly the advantage that it can bring to the task of visual attention prediction. As stated in Sec. 4.2: "for this method, the priority map is created by applying a Gaussian filter to a binary map where the center of the bounding box over the target(s) is set to 1 and everywhere else 0", this implies that the points in the binary map with values==1 (of high priority) are mostly located at the center of the bounding box of each object, and, as can be observed in the figures showing animal-related classes, this is not the same as the fixation points. This is further confirmed in Table 1, where it can be noted that is in the animal-related classes (cat, cow, dog and horse) where the proposed SDR method has a significant advantage over AnnoBoxes (with boosts of 5, 4, 8 and 6 percentage points, respectively). In my opinion, a comparison against the two methods listed above (R-CNN and AnnoBoxes) is not a strong empirical evidence that the proposed method is good at predicting visual attention (fixation points). Perhaps, from the results of the previous two comparisons, it is better to say that solely focusing on learning how to localize objects does not imply learning how to predict visual attention (fixation points). This could serve as a motivation for the diversity introduced by SDR via NMS. I believe the paper can be strengthened by performing experiments to evaluate the effect that some of the parameters (e.g., overlap considered when doing NMS for SDR, width of the Gaussian blur kernel) have on performance. In addition, I suggest making more concrete the conclusions made during this work (At this point they were not so clear to me). In addition, regarding to the comparisons wrt. to existing work, I suggest considering the method from Li et al. arXiv:1506.06343 which tries to detected visual patterns on image regions that can be informative for image classification. In my opinion this method has some similarities wrt. the method proposed in the paper. In my opinion the paper is still at a premature stage for publication at this point . Given the promising potential and links that the proposed method has in different fields, I encourage the authors to properly position their work wrt. existing work, revise the baselines considered during their evaluation, and, through an ablation study, provide an insight on the effect of different parameters that are part of the proposed method.

Confidence in this Review

2-Confident (read it all; understood it all reasonably well)


Reviewer 2

Summary

This paper suggests that the sparse and diverse features learned by RRSVM also predicts visual saliency. Experiments were designed to show the correctness of this hypothesis.

Qualitative Assessment

This paper suggests the sparse and diverse features learned by RRSVM for classification also predicts visual saliency. This idea is interesting and worth in-depth investigation. The paper is well organized and well written. The main drawback is that the authors have not provide significant enough evidence to show the proposed method is better than the simple center bias model. Providing following experiments and comparison could made the paper on a more solid foundation. 1. Use the shuffled AUC (sAUC) criterion [R1, R2] for performance comparison, which is designed for handling center bias. 2. Comparisons with the methods in [R3, R4] could provide stronger evidence. 3. Another way to improve the criterion is to compare the positions of selected regions directly with the eye fixation points with the metric introduced in [R5]. [R1] L. Zhang, M. H. Tong, T. K. Marks, H. Shan, and G. W. Cottrell, "SUN: A Bayesian framework for saliency using natural statistics,” J. Vis., vol. 8, no. 7, pp. 1–20, Dec. 2008. [R2] B. W. Tatler, "The central fixation bias in scene viewing: Selecting an optimal viewing position independently of motor biases and image feature distributions," J. Vis., vol. 7, no. 14, pp. 1–17, 2007. [R3] A. Garcia-Diaz, V. Leborán, X. R. Fdez-Vidal, and X. M. Pardo, "On the relationship between optical variability, visual saliency, and eye fixations: A computational approach," J. Vis., vol. 12, no. 6, pp. 1–22, 2012. [R4] C. Xia, F. Qi, G. Shi, "Bottom-up Visual Saliency Estimation with Deep Autoencoder-based Sparse Reconstruction," IEEE Trans. Neural Networks and Learning Systems, 27(6): 1227–1240, June 2016. [R5] A. Borji, H. R. Tavakoli, D. N. Sihite, and L. Itti, "Analysis of scores, datasets, and models in visual saliency prediction," in Proc. 14th IEEE Int. Conf. Comput. Vis., Sydney, VIC, Australia, Dec. 2013, pp. 921–928.

Confidence in this Review

3-Expert (read the paper in detail, know the area, quite certain of my opinion)


Reviewer 3

Summary

The authors augment a model of image classification, RRSVM, with a simple diversity mechanism, and achieve state-of-the-art results in predicting the location of human gaze fixations during visual search.

Qualitative Assessment

Preface: I am not a vision researcher (either behavioral or computational). I think this paper is an accept -- the premise of using a classification model for predicting search fixations is clever, this achieves state of the art performance, and the work is clearly presented. Some suggestions for improvement: 1. What is the split-half AUC for the human data? If the models have higher AUC than this, it's probably a sign that you should collect more data. Conversely, the human AUC can provide a natural upper bound on model AUC -- assuming you have a reliable estimate of human AUC, it would be hard to fault you if your model AUC didn't exceed this. 2. Can you provide a measure of uncertainty for the mean AUCs of RRSVM and SDR? e.g., run these with different validation sets and report standard error of the mean; it would be nice to show that the 0.85 score of SDR is significantly different from the 0.81 score of RRSVM. 3. Out of curiosity: I take it on faith that AUC is the most widely used metric for fixation prediction, but why doesn't the field use something like log-likelihood of the human data under the priority map? 4. The writing in section 2 strikes me as pretty similar to section 3.1 of Wei and Hoai's original CVPR paper on RRSVM. If you are not actually Wei and Hoai, consider revising the language to be more distinctive. 5. As a non-computer-vision person, it would be helpful to know exactly what regions RRSVM searches over. Looking at Wei and Hoai, the regions appear to be rectangular but it's not clear how large/small they can be, how many possible regions there are, etc. Also, it would be helpful to know how much the SDR region selection function Psi reduces the dimensionality of the region space. 6. Table 1 shows that all the models take a performance hit on the and sofa items. Looking at Wei and Hoai, it appears that base RRSVM also takes a performance hit in classification -- is there something to say here about the relationship between classification and fixation prediction? (e.g., is good classification prediction necessary for good fixation prediction?) ----- Update: I read the other reviews and the rebuttal, which seems to adequately address the technical issues raised by reviewers, so I am still in favor of this paper.

Confidence in this Review

2-Confident (read it all; understood it all reasonably well)


Reviewer 4

Summary

The paper proposes a visual attentional methods based on a RRSVM classifier. RRSVM recognizes image category using a sparse set of regions. The visual features is extracted using a deep neural network (VGG16). To detect multi-objects in an image, a biological mechanism IOR is added into the optimization objective of the RRSVM to encourage diversity during training. The method achieves competitive results on visual serach tasks of single object and multi objects.

Qualitative Assessment

The IOR adopted by the paper is reasonable, but sensitivity analysis on IOR related parameters should be provided as the supression area will affect the prediction accuracy of multi-objects at different scales. This may also explain some failure examples as shown in Figure 3. Another issue is about the experiment in Table 1. The proposed SDR is basded on a VGG16 as a feature extractor and a classifier RRSVM, and the novelty of this paper lies in the classifier RRSVM instead of feature extractor. Therefore, the experiments should be carried on by using the same feature extractor across all methods. However, RCNN and CAM use different feature extractors. So the contribution of RRSVM is not clear.

Confidence in this Review

2-Confident (read it all; understood it all reasonably well)


Reviewer 5

Summary

The paper adds to the region sparsity model an inhibition of return mechanism, and investigates the relationship of the new model with visual attention.

Qualitative Assessment

Studies on sparsity and attention have been there for a while, e.g., Hou, Harel, and Koch (PAMI 2012). The authors claim the link of the two in the manuscript, yet failed to provide theoretical justifications, thus the link appears somewhat superficial in this work. It is a good idea to consider inhibition of return in attention models, yet the current addition looks ad-hoc. It would help if the authors could elaborate this part, with mathematics or experiments. For example, for the latter, I would expect experimental results showing with and without this mechanism for a direct comparison and demonstration of its effectiveness. Sections 2 and 3 are quite disconnected from the rest. In fact they look similar in form with ref [29]. I would suggest the authors to reformat and to fit in the current context, i.e., visual attention. The authors design and report several task paradigms, which are nice. The qualitative figures, however, did not explicitly show the strength of the new approach in these contexts. Test stimuli are simple, say with one or two objects in relatively isolated scenes. It is not clear whether the accuracy is achieved due to straightforward factors like they are object regions, or they are large objects. I would expect experiemental demonstrations on more complex scenes, and a more thorough discussion on this. I would change the visualization of the maps, the current heatmap blocks the original stimuli and the human fixations to quite a large extent.

Confidence in this Review

3-Expert (read the paper in detail, know the area, quite certain of my opinion)


Reviewer 6

Summary

This paper introduces a technique which modifies the Region Ranking SVM to become a biologically plausible visual attention attention model. Given that the Region Ranking SVM puts emphasis on a small amount of non-uniform regions of visual space to do its classification, it makes sense that those areas are most salient for classification. By choosing an fixation order based on the RRSVM and adding an inhibition of return mechanism, it allows visual search to maximally attend to areas of importance while decreasing the likelihood of attending to nearby already-attended areas. The results show physiologically-consistent behaviour and well as better or comparable results to other similar techniques. It also presents situations where the algorithm does not behave as well, which is very good to understand the benefits and limitations of this work such that future work can use this as a foundation.

Qualitative Assessment

The paper is well-written and the experiments are useful, thorough, and well-explained. In terms of novelty, the RRSVM does the bulk of the work here and it is a straightforward progression to use the most important areas as fixation points for visual attention. However, the use of inhibition of return and positioning the work in a physiological domain (including the results) makes the work more valuable than just using the RRSVM. The lack of biological plausibility in the implementation of the RRSVM might be a reason that the entire framework is not investigated further for a more direct understanding of biological visual search. Ignoring the specifics of the implementation and the incremental nature of the work, and focusing on the concept and experimental method/results, this is useful to shine some light on biological visual attention.

Confidence in this Review

2-Confident (read it all; understood it all reasonably well)